# Latest Results from the T2K Neutrino Experiment[†]

**Dean Karlen and on behalf of the T2K Collaboration**

Department of Physics and Astronomy, Univeristy of Victoria and TRIUMF, Victoria, BC V8P 5C2, Canada;
karlen@uvic.ca

**Abstract:** The T2K long baseline neutrino oscillation experiment measures muon neutrino disappearance and electron neutrino appearance in accelerator-produced neutrino and anti-neutrino beams. This presentation reports on the analysis of our data from an exposure of $2.6 \times 10^{21}$ protons on target. Results for oscillation parameters, including the CP violation parameter and neutrino mass ordering, are shown.

**Keywords:** neutrino; neutrino oscillation; neutrino properties; CP violation

## 1. Introduction to Lepton Mixing

Super-Kamiokande [1] and Sudbury Neutrino Observatory [2] measurements of neutrinos produced naturally in the sun and from interactions of cosmic rays with the atmosphere established that leptons mix and that neutrinos have mass. These achievements led to the 2015 Novel Prize in Physics being awarded to Takaaki Kajita and Arthur B. McDonald.

Lepton mixing is described by the Pontecorvo–Maki–Nakagawa–Sakata (PMNS) matrix, $U$, such that each neutrino flavour state, $|\nu_\alpha\rangle$, is a linear combination of mass states, $|\nu_i\rangle$, with mass $m_i$,

$$|\nu_\alpha\rangle = \sum_i U_{\alpha i}^* |\nu_i\rangle . \tag{1}$$

Due to this mixing, a neutrino produced with a definite flavour $\alpha$ can be detected as another flavour $\beta$. In the absence of matter, the amplitude for flavour change of a neutrino with energy $E$ a distance $L$ from its production point is given by [3]

$$\text{Amp}\left(\nu_\alpha \to \nu_\beta\right) = \sum_i U_{\alpha i}^* e^{-im_i^2 L/(2E)} U_{\beta i} . \tag{2}$$

The corresponding probability for neutrino flavour change is

$$
\begin{aligned}
P\left(\nu_\alpha \to \nu_\beta\right) &= \left|\text{Amp}\left(\nu_\alpha \to \nu_\beta\right)\right|^2 \\
&= \delta_{\alpha\beta} - 4 \sum_{i>j} \text{Re}\left(U_{\alpha i}^* U_{\beta i} U_{\alpha j} U_{\beta j}^*\right) \sin^2\left(\Delta m_{ij}^2 \frac{L}{4E}\right) \\
&\quad + 2 \sum_{i>j} \text{Im}\left(U_{\alpha i}^* U_{\beta i} U_{\alpha j} U_{\beta j}^*\right) \sin^2\left(\Delta m_{ij}^2 \frac{L}{2E}\right)
\end{aligned}
\tag{3}
$$

where $\Delta m_{ij}^2 \equiv m_i^2 - m_j^2$. The observation of a neutrino flavour change, also referred to as neutrino oscillation, therefore implies non-degenerate neutrino masses. Two mass separations were found, the solar separation being $\Delta m_{21}^2 \approx 7 \times 10^{-5}$ eV$^2$ and the atmospheric separation $\left|\Delta m_{32}^2\right| \approx 2 \times 10^{-3}$ eV$^2$. It is not known whether the overall ordering of the neutrino masses is "normal" (with the larger

separation between the two highest mass neutrino mass states) or the alternative, referred to as "inverted" mass ordering.

The PMNS matrix is usually parameterized in terms of the mixing angles $\theta_{ij}$ and CP violation phase $\delta_{CP}$:

$$
U = \begin{bmatrix} 1 & 0 & 0 \\ 0 & c_{23} & s_{23} \\ 0 & -s_{23} & c_{23} \end{bmatrix} \begin{bmatrix} c_{13} & 0 & s_{13}e^{-i\delta_{CP}} \\ 0 & 1 & 0 \\ -s_{13}e^{i\delta_{CP}} & 0 & c_{13} \end{bmatrix} \begin{bmatrix} c_{12} & s_{12} & 0 \\ -s_{12} & c_{12} & 0 \\ 0 & 0 & 1 \end{bmatrix} , \tag{4}
$$

where $c_{ij} = \cos\theta_{ij}$ and $s_{ij} = \sin\theta_{ij}$.

## 2. Experiments with Artificial Neutrinos

The Super-Kamiokande and Sudbury Neutrino Observatory measured the atmospheric and solar mixing angles ($\theta_{23}$ and $\theta_{12}$, respectively) with natural neutrinos. In the past decade, experiments with artificial neutrinos have improved lepton mixing measurements and established that lepton mixing is complete, in that $\sin\theta_{13} \neq 0$. In particular, the Daya Bay, Reno, and Double Chooz experiments used neutrinos produced at nuclear reactors and the K2K, T2K, Minos, and NOvA experiments used neutrino beams produced from proton accelerators. It is not yet established whether the mass ordering is normal or inverted and whether there is CP violation in neutrino oscillations. If $\sin\delta_{CP} \neq 0$, then $P\left(\nu_\alpha \to \nu_\beta\right) \neq P\left(\bar{\nu}_\alpha \to \bar{\nu}_\beta\right)$.

An idealized experiment offering the simplest analysis and smallest systematic uncertainty would have a parallel beam of mono-energetic neutrinos of one flavour directed toward two separated identical detectors having unambiguous charged-lepton identification. The mixing parameters would be determined by measuring the oscillation probability for different separation distances and/or neutrino energies. The oscillation probability would be estimated by correcting the ratio of the number of neutrino events producing a $\beta$ type charged lepton in the far detector to the number producing an $\alpha$ type charged lepton in the near detector,

$$
p^{est}\left(\nu_\alpha \to \nu_\beta \mid L, E\right) = \frac{N_\beta^{far}}{N_\alpha^{near}} \times \frac{\phi^{near}}{\phi^{far}} \frac{\sigma_\alpha}{\sigma_\beta} \frac{\varepsilon_\alpha}{\varepsilon_\beta} = \frac{N_\beta^{far}}{N_\alpha^{near}} \times \frac{\sigma_\alpha}{\sigma_\beta} \frac{\varepsilon_\alpha}{\varepsilon_\beta} \tag{5}
$$

With perfectly parallel beams, there is no uncertainty arising from neutrino flux ($\phi$) and with identical detectors the remaining systematic uncertainties arise from differences in neutrino cross sections ($\sigma$) and detector efficiencies ($\varepsilon$) for flavours $\alpha$ and $\beta$.

Unfortunately, creating a parallel beam of neutrinos is not possible since after their production, their direction cannot be controlled. For the long distances required for substantial oscillation probability of muon neutrinos, the near/far flux ratio can be of order $10^6$, which may necessitate different detector designs. Also, mono-energetic neutrino beams are not possible in general, and therefore it is necessary to accurately model the neutrino spectrum and estimate the energy of each interacting neutrino. T2K reduces the systematics related to both of these issues, by using a narrow band neutrino beam at energies where the neutrino energy is well estimated by the charged lepton momentum and angle alone. While the idealized experiment is not realistic, most oscillation experiments incorporate near and far detectors to reduce systematic uncertainty. With this approach, it is only necessary to model relative (rather than absolute) fluxes, cross sections, and efficiencies between the two detectors.

## 3. The T2K Experiment

The T2K experiment [4] extracts 30 GeV protons from the J-PARC main ring to strike a long graphite target to produce hadrons. Charged hadrons, sign-selected by magnetic horns surrounding and downstream of the target, decay in flight within a decay pipe to produce a beam of predominantly muon neutrinos (or muon anti-neutrinos) towards the Super-Kamiokande detector, approximately 295 km away.

A complex of near detectors, 280 m downstream of the target, measures neutrino properties prior to oscillation and the far detector (Super-Kamiokande) directly measures the effect of lepton mixing.

The neutrino beam axis is directed 2.5° away from the far detector to optimize the sensitivity to oscillation parameters for the 295 km baseline. Figure 1 shows the spectrum of neutrinos passing through the far detector overlayed with the $\nu_\mu$ survival and $\nu_e$ appearance probabilities for representative oscillation parameters.

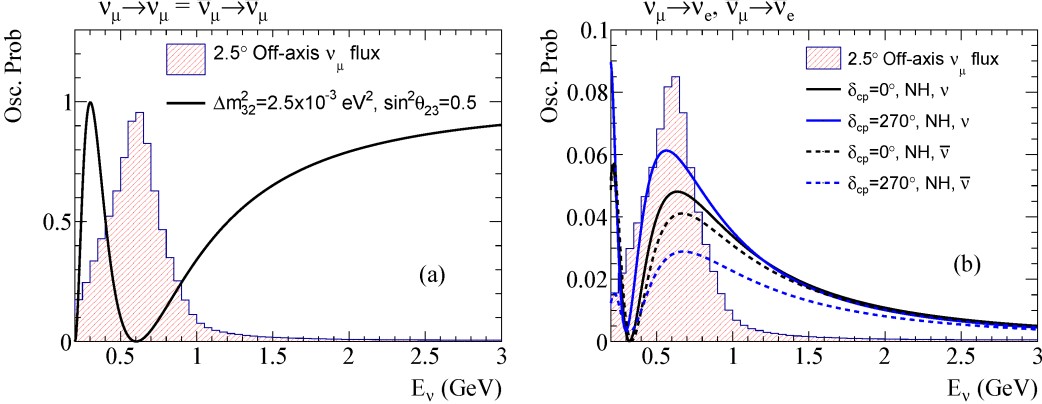

**Figure 1.** The spectrum of neutrinos passing through the T2K far detector, peaking at approximately 0.6 GeV due to the 2.5° off axis angle, is compared with the energy dependence of the muon–neutrino oscillation probabilities. (**a**) The muon–neutrino survival probability is near zero at the peak of the neutrino spectrum. (**b**) The electron–neutrino appearance probability is near maximum at the peak. The different curves correspond to different CP violation parameters for normal mass ordering for neutrinos and anti-neutrinos. The difference between neutrino and anti-neutrino oscillation probability for $\delta_{CP} = 0°$ gives an indication of T2K sensitivity to mass ordering, while the difference for $\delta_{CP} = 270°$ shows the much larger effect that would arise with maximum CP violation.

The T2K near detector complex consists of the INGRID on-axis detector that directly measures the neutrino beam profile with high statistics and the ND280 off-axis detector that measures neutrino interactions in greater detail using a set of sophisticated detector systems that form a magnetized spectrometer. The ND280 detector uses the magnet built for the UA1 experiment and samples the neutrino beam travelling in the direction towards the far detector. The oscillation analyses use events, like the one shown in Figure 2a, that arise from neutrino interactions in the two Fine Grained Detectors, containing many planes of narrow scintillator bars and water layers. The resulting charged particles passing through the neighboring Time Projection Chambers have their charges and momenta determined by tracking measurements, and their particle types identified by measurements of their ionization. Surrounding calorimeters and range detectors provide additional information. The event rates and kinematics are used to tune models for the neutrino flux and neutrino interactions.

The T2K far detector, Super-Kamiokande, is a cylindrical underground cavern containing 50,000 tons of pure water surrounded by 11,000 20-inch photomultiplier tubes (PMTs). Charged particles passing through the water at speeds exceeding that of light in the water emit Cerenkov radiation. Events corresponding to interactions of T2K neutrinos are selected by considering only those arriving during the narrow time windows around the neutrino bursts produced by the J-PARC proton beam, and by requiring that the produced charged particles start and end within the fiducial volume. Such events are identified as rings of light on the array of PMTs and the rings produced by electrons are much less sharp due to multiple scattering and showering, than those produced by muons. This provides a very powerful method to distinguish the interactions of electron and muon neutrinos. An event display for an electron–neutrino interaction is shown in Figure 2b.

T2K began collecting data in 2010. The beam power has steadily increased and recently 500 kW has been achieved. In total, T2K has collected $3.16 \times 10^{21}$ protons on target, split roughly equally between neutrino and anti-neutrino modes.

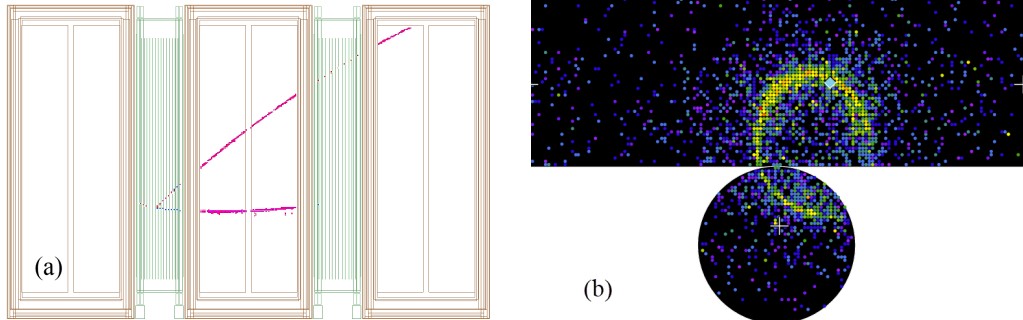

**Figure 2.** Two event displays for neutrino interactions observed by the (**a**) near and (**b**) far detectors of T2K. (**a**) A charged current muon–neutrino interaction in the first Fine Grained Detector produces tracks in the two downstream Time Projection Chambers; (**b**) An energetic electron from a neutrino interaction inside the Super-Kamiokande detector generates Cerenkov light which is detected by photomultipliers in a cylindrical structure. The display shows the pattern of light on the wall (unwrapped) and the base.

## 4. T2K Results

The near detector has collected a large number of neutrino interaction events which are used to test and refine neutrino interaction models. As one example, Figure 3 illustrates the selection of charged current (CC) muon–neutrino interactions that also produce a $\pi^0$, and the spectrum of muons is shown, along with the expected spectra from various interaction types as predicted by the NEUT model. The overall cross section is in agreement with the model,

$$\frac{\sigma_{Data}}{\sigma_{NEUT}} = 1.18 \pm 0.03 \ (stat) \ ^{+0.22}_{-0.21} \ (sys). \tag{6}$$

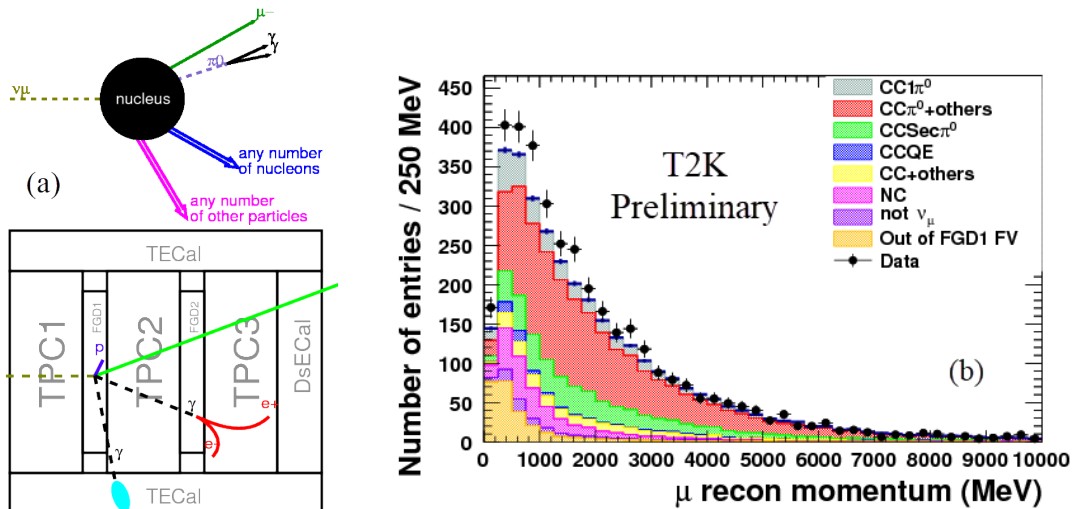

**Figure 3.** $\pi^0$ production in muon–neutrino charged current interactions is measured by the T2K near detector, ND280. (**a**) An illustration of the criteria applied to select a sample enhanced with CC$\pi^0$ events and a typical topolgy of these events in ND280; (**b**) Observed $\mu^-$ spectrum (points) compared to the NEUT model predictions broken down by interaction category.

### 4.1. Neutrino Oscillation Analyses

To estimate the neutrino oscillation parameters described in Section 1, we use models for the neutrino flux and neutrino interactions, as well as for the performance of the near and far detectors. The models

include systematic parameters to encapsulate our uncertainty, both theoretical and experimental. Some of the systematic parameters are constrained using external data, for example hadron production measurements from the NA61 experiment. We measure kinematic distributions of the leptons contained in samples enhanced in different neutrino interaction types in the near and far detectors to form likelihood functions. We use these likelihood functions for both frequentist and Bayesian interpretations for the physics parameters while marginalizing over the systematic parameters.

Figure 4 illustrates the power of the near detector data by comparing the observed muon spectrum in quasi-elastic charged current interactions with our models before and after optimization of the systematic parameters. In the oscillation analyses, the systematic parameters are treated as random variables that are constrained by such data; they are not simply fixed to their optimal values. Both the lepton momenta and the directions of the leptons, with respect to the neutrino beam axis, are treated in this manner. These two observables are sufficient to form a good estimate of the neutrino energy for quasi-elastic events.

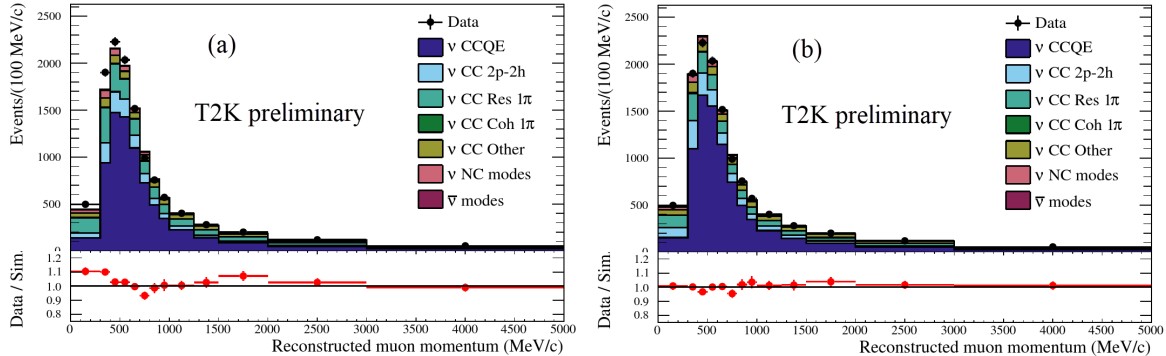

**Figure 4.** The muon spectrum in ND280 event samples enhanced in quasi-elastic charged current interactions is compared to the model expectations for the (**a**) nominal systematic parameters and (**b**) optimized systematic parameters. The optimization takes into account constraints from external data.

### 4.2. Neutrino Oscillation Results

Substantial disappearance of muon–neutrinos (and muon anti-neutrinos) is observed by the far detector as shown in Figure 5 which also shows the T2K confidence intervals for the atmospheric oscillation parameters.

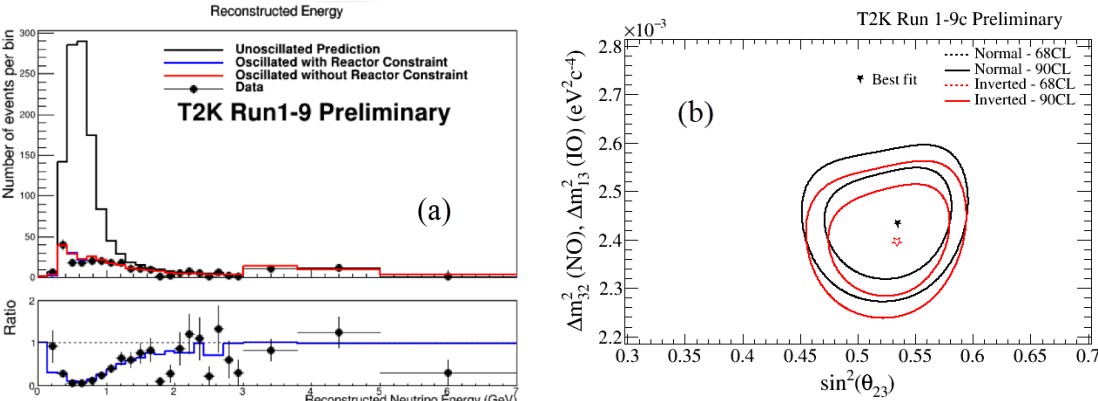

**Figure 5.** The rate of muon–neutrinos observed in the far detector is significantly suppressed due to neutrino oscillation. (**a**) The expected rate without neutrino oscillation (unoscillated prediction) is compared to the observed rate. Also shown are expected rates for the best fit oscillation parameters. There is no observable difference if reactor data is included in the fit. (**b**) Confidence intervals for the atmospheric oscillation parameters are shown separately for the normal and inverted mass ordering hypotheses.

The rate of (anti) electron–neutrino appearance in the primarily (anti) muon–neutrino beam depends on the CP-violation parameter $\delta_{CP}$ and the mass ordering (*MO*), neither of which are known. As a result, these measurements command a great deal of interest in the neutrino physics community. With the current dataset, T2K has only limited sensitivity to these physics parameters, as shown in Figure 6. The physics parameters of interest, *CP* and *MO*, have a similar effect on the expected event rates. If the T2K observation was within the central part of the range of expectations, predicted by both normal and inverted mass ordering, that data would not show a preference between the two *MO* as the likelihood ratio would be 1. Instead, the T2K observation is outside the range of expected rates and the likelihood of observing this data is greatest for normal mass ordering and $\delta_{CP} = -\pi/2$.

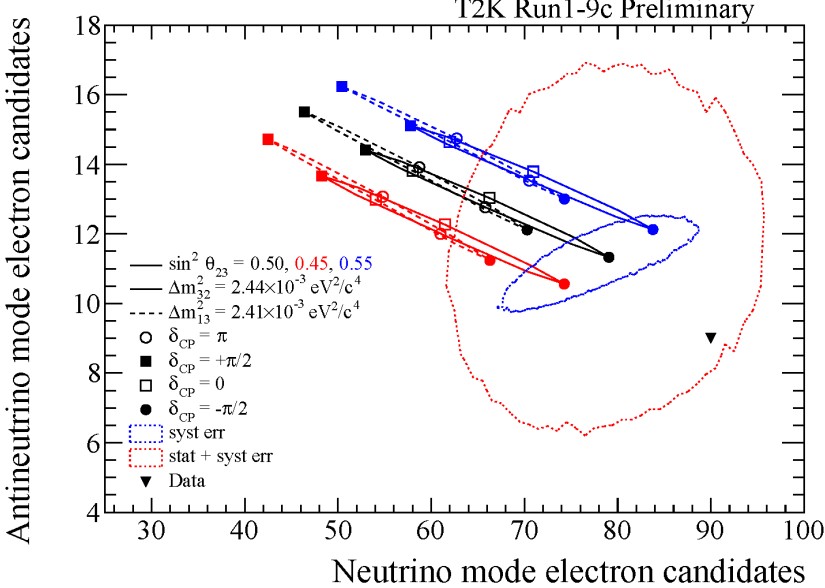

**Figure 6.** The three pairs of narrow ellipses show the expected numbers of anti electron–neutrino events and electron–neutrino events for optimized systematic parameter values. The solid (dashed) ellipses are for normal (inverted) mass ordering. The value of $\delta_{CP}$ locates the prediction on any of the ellipses. The observed number of events is shown by the inverted triangle. The region with the blue jagged border centred around the expectation for ($\sin^2\theta_{23} = 0.5$, $\delta_{CP} = -\pi/2$, normal ordering) shows the region containing 68% of the predictions (for that choice of physics parameters) when systematic parameters are treated as random variables constrained by T2K and external data. The larger region with the red jagged border represents 68% of experimental outcomes distributed according to Poisson random variables whose expectation values are distributed due to the systematic uncertainties.

The T2K sensitivity to both *CP* and *MO* is strongly dependent on the true values. This is demonstrated by calculating the posterior probability density for $\sin\delta_{CP}$ and the posterior odds for normal ordering for hypothetical T2K experiments that collected the same number of protons on target but observed exactly the expected number of events. The event rates are approximately linear in the parameter $\sin\delta_{CP}$ and therefore the posterior densities are similar to Gaussian distributions truncated to the allowed range for that parameter. Figure 7 shows that the posterior probability distribution and posterior odds are dramatically different for the two sets of physics parameters considered, and also differ to that of the actual T2K observation. Also shown in the figure is the likelihood ratio for the parameter $\delta_{CP}$ and the resulting frequentist confidence interval. The *CP* conserving values $\sin\delta_{CP} = 0$ are excluded from the $2\sigma$ credible and confidence intervals. Normal ordering is strongly preferred.

Figure 7 shows that the current T2K credible interval on $\sin\delta_{CP}$ is smaller than expected, which is due to the fact that the observed rates fall outside of the region of expected values. If future T2K data brings the rates closer to the expected values, it is possible that the credible and confidence intervals on $\delta_{CP}$ may not reduce in size and the posterior probability for normal ordering may not increase.

It is also important to not overinterpret the T2K result as providing strong evidence for maximal *CP* violation, as there is a large range of values of $\delta_{CP}$ contained in the credible and confidence intervals. Any hypothetical T2K outcome with the current protons on target that excluded $\sin \delta_{CP} = 0$ at $2\sigma$ would necessarily also find that maximal CP violation has the largest likelihood.

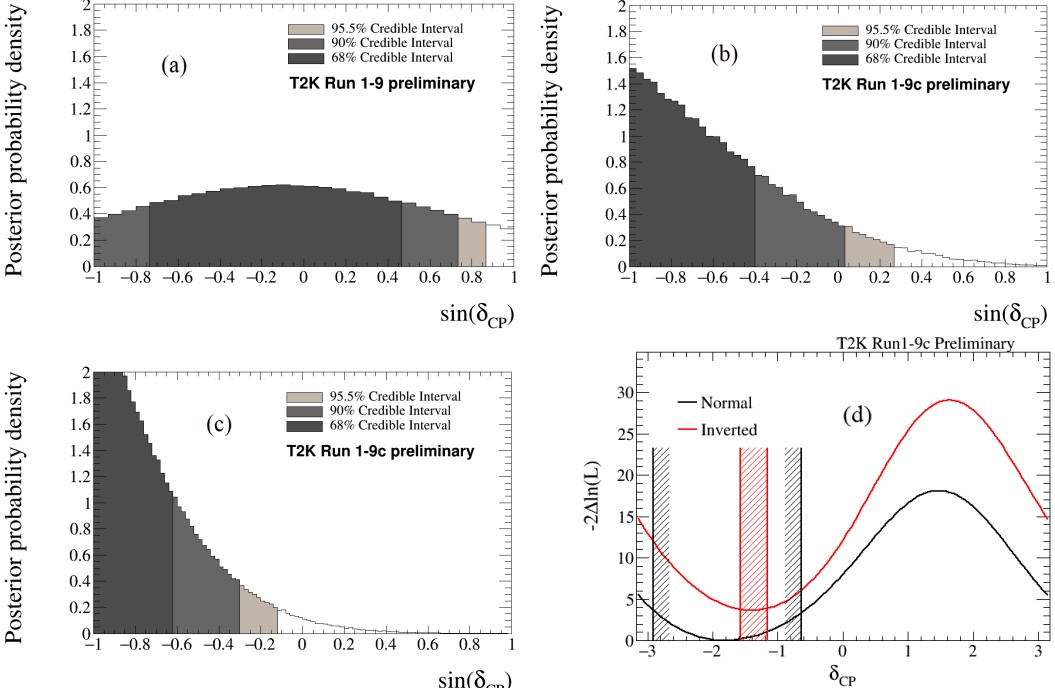

**Figure 7.** The posterior probability density for $\sin \delta_{CP}$ marginalized over all other parameters including mass ordering (**a–c**). The shaded regions show 68%, 90%, and 95.5% credible intervals. (**a**) The outcome for an experiment observing the expected number of events for ($\sin^2 \theta_{23} = 0.45$, $\delta_{CP} = 0$, normal ordering). The posterior odds is 1:1 (not favoring either mass ordering); (**b**) The outcome for an experiment observing the expected number of events for ($\sin^2 \theta_{23} = 0.53$, $\delta_{CP} = -\pi/2$, normal ordering). The posterior odds are 2.7 in favor of normal ordering; (**c**) The outcome for the T2K observation. The posterior odds are 7.9 in favor of normal ordering and $\sin \delta_{CP} = 0$ is outside of the 95.5% credible interval; (**d**) The frequentist $2\sigma$ confidence intervals on $\delta_{CP}$ (selected using the likelihood ratio) shown separately for the two mass orderings.

## 5. Conclusions and Outlook

The T2K experiment has been very successful in furthering our understanding of neutrino oscillation. Early in its experimental program, T2K was the first experiment to give an indication for $\theta_{13} \neq 0$ which opened the door for exploring *CP* violation in long baseline neutrino oscillation experiments [5].

The T2K $2\sigma$ confidence and credible intervals exclude the CP conserving value $\delta_{CP} = 0$, suggesting that CP is violated in the lepton sector. Normal mass ordering is preferred over the inverted mass ordering by a factor of 7.9:1. Recent results from NOvA also favor normal mass ordering, but for that mass ordering disfavor $\delta_{CP} = -\pi/2$ (the value with the largest likelihood for T2K data) [6]. It is important to correctly account for the dependence between the estimators for mass ordering and $\delta_{CP}$ for any combination of T2K and NOvA data. In particular, marginalizing over $\delta_{CP}$ must not be done independently when calculating the joint posterior probability for normal ordering.

Further improvement in understanding *CP* and *MO* will require more data from NOvA and a potential run extension of T2K, with JPARC continuing to ramp up beam power. Greater advances will come when the future programs, HyperK and DUNE become operational in the next decade.

**Conflicts of Interest:** The authors declare no conflict of interest.

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
