# Peer review of "Latest Results from the T2K Neutrino Experiment"

_universe, doi:10.3390/universe5010021_

Round 1

Reviewer 1 Report

This work is clearly interesting and relevant for readers involved in the topics of neutrino physics. It deserves publication.

I've just a doubt. Giving a look at the most recent literature of the author, I noticed that there exists a paper on arxiv: 1807.07891 which discusses similar things. 

Which are the main differences with respect to this paper? Moreover if parts are taken from the aforementioned paper, the author must account for changing such parts, giving an original report on the topic.

After these changes the paper can be accepted.

Author Response

Thank you for the comment/question. I address the point you raised below:

I've just a doubt. Giving a look at the most recent literature of the author, I noticed that there exists a paper on arxiv: 1807.07891 which discusses similar things.

Which are the main differences with respect to this paper? Moreover if parts are taken from the aforementioned paper, the author must account for changing such parts, giving an original report on the topic.

The conference report provides results from an updated analysis, with approximately 50% more data collected in anti-neutrino mode, compared to the arxiv article.

The conference report is written for a general science audience, in contrast to the arxiv article written primarily for experts.

No text from the arxiv article was copied into the conference report. My report is an original manuscript. The conference report draft was circulated within the T2K collaboration for comment, prior to submitting to this journal.

I hope this clarifies the situation with the reviewer. I don't believe that any changes to the manuscript are necessary.

Reviewer 2 Report

In this paper the author reports the latest results from the T2K neutrino experiment on behalf of the T2K Collaboration.

The research design is appropriate and the methods and results are adequately and clearly presented.

The author is english native tongue and there are no grammar error.

To my opinion the paper have to be accepted in the present form.

Author Response

Thank you for reviewing my submission.

The reviewer does not request changes to the manuscript.

Reviewer 3 Report

This article describes very well the T2K experiment and its results.

I recommend its publication.

I only suggest to improve the text in Figures 7a and 7b, which are simulations of an hypothetical experiment.
The text "T2K Run 1-9 preliminary" is confusing.

Author Response

Thank you for reviewing this manuscript.

One point was raised:

I only suggest to improve the text in Figures 7a and 7b, which are simulations of an hypothetical experiment.The text "T2K Run 1-9 preliminary" is confusing.

It is true that the credible intervals represent outcomes from hypothetical experiments. Our collaboration requires us to include "T2K Run 1-9 preliminary" for these figures. The reason is that the intervals also depend on the assessment of systematic uncertainty of the experiment, which at this time, is still preliminary. It is not unusual to have such designations on figures showing analyses of simulated data in this situation.

I hope this answers the point raised by the reviewer, and I suggest that no change to the manuscript is needed. The T2K figures are "official" and cannot be altered.